# Cardioprotective Agents for the Primary Prevention of Trastuzumab-Associated Cardiotoxicity: A Systematic Review and Meta-Analysis

**DOI:** 10.3390/ph16070983

**Published:** 2023-07-09

**Authors:** Kyriakos Goulas, Dimitrios Farmakis, Anastasia Constantinidou, Nikolaos P. E. Kadoglou

**Affiliations:** Medical School, University of Cyprus, Nicosia 2029, Cyprusconstantinidou.anastasia@ucy.ac.cy (A.C.);

**Keywords:** cardiotoxicity, trastuzumab, trastuzumab-induced cardiotoxicity (TIC), primary prevention, beta-blockers (BBs), angiotensin receptor blockers (ARBs), angiotensin converting enzyme inhibitors (ACEIs)

## Abstract

There are significant considerations about the prevention of cardiotoxicity caused by trastuzumab therapy in patients with breast cancer, leading to discontinuation. Recently, randomized controlled trials (RCTs) have evaluated the effects of early commitment of beta-blockers (BBs), angiotensin receptor blockers (ARBs) and angiotensin converting enzyme inhibitors (ACEIs) during trastuzumab chemotherapy in order to prevent the related cardiotoxicity. The present systematic review and meta-analysis of six RCTs included patients who have predominantly non-metastatic, HER2-positive, breast cancer and received trastuzumab as primary or adjuvant therapy. Those patients did not have any obvious cardiac dysfunction or any previous therapy with cardioprotective agent. We evaluated the efficacy of the aforementioned medications for primary prevention of cardiotoxicity, using random effects models. Any preventive treatment did not reduce cardiotoxicity occurrence compared to controls (Odds ratios (OR) = 0.92, 95% CI 0.54–1.56, *p* = 0.75). Results were similar for ACEIs/ARBs and beta-blockers. Treatment with ACEIs/ARBs led to a slight, but significant, increase in LVEF in patients compared to the placebo group. Only two studies reported less likelihood of discontinuation of trastuzumab treatment. More adequately powered RCTs are needed to determine the efficacy of routine prophylactic therapy.

## 1. Introduction

Breast cancer is one of the most commonly diagnosed cancers in women [1,2]. It is estimated that 15–20% of human breast cancers may overexpress a molecule called Human epidermal growth factor receptor-2 (HER-2) oncogene, which is associated with poor prognosis and a high recurrence rate [3,4,5,6,7]. These “HER-2 positive” cancers are amenable to treatment with targeted biological agents, such as trastuzumab [8,9,10]. The administration of trastuzumab has led to a 30–40% relative improvement in overall survival and these patients can now expect a 10-year survival of up to 85% [11,12]. Among the most significant adverse effects of trastuzumab is cardiotoxicity, which remains the main reasons of trastuzumab discontinuation [3,10,13,14]. Cardiotoxicity is defined as a significant reduction in the left ventricular ejection fraction (LVEF), with or without development of clinical heart failure [15]. The incidence of trastuzumab-induced cardiotoxicity may range between 5 and 11% of treated patients [16,17], but there are no available indices which could reliably predict it. Those number may differ in clinical practice depending on the cohorts, co-morbidities or the way of monitoring. Current guidelines suggest repeated evaluation of LVEF every 3 months during trastuzumab therapy to detect echocardiographic or clinical signs of cardiotoxicity and thereby guide the continuation of therapy or not [15,18].

According to the latest meta-analyses and the 2022 ESC guidelines [15] for the treatment of cardiotoxicity, the most studied and effective agents (except dexrazoxane) remain angiotensin converting enzyme inhibitors (ACEIs), angiotensin II receptor blocker (ARBs) and beta blockers (BBs) [19,20,21]. Accumulated number of small and heterogeneous studies have implicated that neurohormonal inhibitors slow the rate of cardiac dysfunction in patients receiving the classical chemotherapeutic agents like anthracyclines and/or human epidermal growth factor receptor 2 inhibitors [22]. The efficacy of those agents in the treatment of HFrEF has raised the hypothesis that primary prevention of cardiotoxicity in patients with breast cancer, without chemotherapy discontinuation, could be achieved by the early administration of BBs, ACEIs and ARBs even before the development of classical heart failure [23]. Such an approach may prevent the subsidiary progression of HFrEF before it become clinically apparent or between echocardiographic exams. Previous meta-analyses have deal with the prevention of cardiotoxicity in breast cancer patients undergoing any type of chemotherapy [19,21,24,25,26,27]. Thereby, they have evaluated the cardio-prevention therapy in breast cancer patients receiving mixed chemotherapy (i.e., anthracyclines and trastuzumab). None of them has focused exclusively on HER-2 positive patients treated with trastuzumab. There are two possible reasons for the limited number of publications on the prevention of trastuzumab-related cardiotoxicity. The first might be the predominant use of anthracyclines and the second the small number of participants in each study.

The present systematic review and meta-analysis has come to cover this gap by assessing the impact of early commitment of ACEIs, ARBs, and BBs on cardiotoxicity occurrence in trastuzumab-receivers with breast cancer and no pre-existing HF.

The primary objective of this systematic review and meta-analysis was to evaluate the primary prevention of cardiotoxicity, via cumulative cardioprotective effects of pharmaceutical agents (BBs, ACEIs or ARBs) versus placebo, in patients with HER-2 positive breast cancer undergoing trastuzumab chemotherapy without any index of HF at the beginning of the therapy. The secondary objective was the comparative evaluation of ACEIs, ARBs and BBs in cardiotoxicity prevention, by assessing: (1) the number of patients who discontinued trastuzumab therapy, (2) the changes in LVEF or global longitudinal strain (GLS) after 3 months of follow-up and (3) the change in cardiac serological biomarkers. The secondary objective was obtained by comparing patients receiving either a single drug or any combination of those cardioprotective drugs with patients receiving placebo.

## 2. Materials and Methods

This systematic review and meta-analysis was performed according to PICO framework and PRISMA 2020 guidelines [28]. RCTs, observational studies, SRs and MAs were studied. Papers presented at conferences or were only in abstract form were excluded. MEDLINE—PubMed, EMBASE and Cochrane Library databases were searched in English for the period 2000 to 2022 using a combination of keywords. An analysis of references of relevant systematic reviews and meta-analyses was conducted.

### 2.1. Search Strategy, Eligibility Criteria and Outcomes

The database review process (MEDLINE—PubMed, EMBASE and Cochrane Library) included studies from 1990 till January 2023. Reference lists cited in systematic reviews and meta-analysis on the same or similar topic were examined.

The search strategy was carried out in the individual databases through the use of keywords. All possible combinations of the following keywords were used: “Cardiotoxicity” “Cardiac dysfunction” “Chemotherapy” “Primary prevention” “Prevention” “Cardioprotective drugs” “Cardioprotective agents” “Cancer” “Breast cancer” “Trastuzumab” “HER2-positive cancer” “beta-blockers (BBs)” “angiotensin receptor blockers (ARBs)” “angiotensin converting enzyme inhibitors (ACEI)”. Combinations of three or more keywords were used with the mandatory use of the words “Cardiotoxicity” and “Prevention” in each search. Filters were used in the search strategy to include only results of studies (type of studies mentioned above) conducted in English.

A predetermined patient-intervention comparison outcome format was used to define our inclusion criteria. The eligibility criteria included studies that had been done in females, diagnosed with HER-2 positive breast cancer, with normal ejection fraction (and GLS if available) at baseline and without HF clinical symptoms or signs. All patients were in the same age group. No patient had received any of the understudied cardioprotective agents prior to trastuzumab chemotherapy. All patients were followed-up for at least three months. The method of comparison between drugs (Drug vs placebo, Drug vs drug, Combination of drugs vs placebo) and the outcomes (prevention, no effect, side effects of cardioprotective agents) should be clearly and comprehensively stated. The primary outcome of interest was cardiotoxicity, as defined by an absolute reduction in LVEF of ≥10% or a reduction in LVEF to <50% from any baseline value [29]. The secondary outcomes of interest were the number of patients who had to discontinue trastuzumab treatment and the changes in measured echocardiographic parameters (LVEF, GLS) and changes observed in serum biomarkers during follow-up.

### 2.2. Data Collection and Extraction

Studies were identified and assess for their eligibility by two independent authors (N.K. and K.G.) using the aforementioned search strategy. The systematic review was carried out independently by the two authors including only RCTs. Any disagreement arosen between the two researchers was resolved by a third reviewer (D.F.) who performed as referee to reach a decision after extensive data review. Extracted data included information pertaining to study type, methodology, population characteristics, interventions, and outcome measures. The database was searched from 2000 to December 2022. The results obtained included primary prevention of cardiotoxicity in HER-2 positive patients treated with trastuzumab. The initial selection of papers was based on the title and abstract of the papers. Subsequently, the full text of the papers was studied and only papers that met the inclusion criteria were selected. In each paper, important elements relating to the results of the study were sought and highlighted. Pre-specified outcome data of interest: (1) primary prevention of cardiotoxicity and the definition of cardiotoxicity, (2) clearly stated that the chemotherapy of HER-2 positive patients was trastuzumab solely or if there was a combination of trastuzumab with other chemotherapeutics, statistical analysis should be reported separately for the trastuzumab arm, (3) cardioprotective agent and the dosage at which it was used, (4) characteristics of the study population, (5) stratification procedure, (6) type of measurements and clinical examination performed on the patients, (7) duration of trastuzumab chemotherapy, (8) duration of follow-up.

### 2.3. Quality Assessment

The study quality was assessed using the GRADE score. The GRADE criteria were applied to the overall analysis of our outcomes in order to grade the certainty of evidence. Six domains of the GRADE criteria were considered: random sequence generation, allocation concealment, blinding, outcome data, reporting bias and other biases. Publication bias was illustrated in a funnel plot and also assessed using Egger’s test. We graphically represented the risk of bias for each study. The studies received the final grade as High, Moderate and Low. The Very low assessed studies excluded from our meta-analysis. Risk of bias and publication risk were independently assessed by the two authors (N.K. and K.G.) using the Cochrane risk-of-bias (RoB2) tool [30]. Disagreements were resolved by the third referee.

### 2.4. Statistical Analysis

All analyses were performed using a random-effects model using the statistical tool of RevMan version 5.4.1. Outcomes of interest included cardiac function as determined by LVEF or GLS, discontinuation of HER2 therapy, and change in cardiac specific biomarkers. For dichotomous outcomes (HER2 treatment discontinuation and number of patients with cardiotoxocity) differences were expressed as odds ratio (OR) with 95% confidence interval (CI) and mean difference (MD) with 95% CI for continuous outcomes (change in LVEF and GLS). The Mantel-Haenszel (M-H) random effects model was used for dichotomous outcomes and the inverse variance random effects model was used for continuous outcomes. To ensure homogeneity in data pooling, studies that reported outcomes consistent with the definition for cardiotoxicity, LVEF measured before and at 3 months follow-up from treatment completion and HER2 treatment discontinuation were considered eligible. The choice of a random-effects model was decided upon a priori due to the expected heterogeneity between different studies (e.g., different populations and different chemotherapeutic regimens). Heterogeneity was assessed via visual inspection of forest plots and I^2^ measure. The I^2^ was interpreted in accordance with recommendations from the Cochrane handbook [31]. A *p*-value of <0.05 for heterogeneity was also deemed likely to reflect a high likelihood of differences beyond chance. Two-sided *p* values <0.05 without correction for multiplicity were considered statistically significant. A trim & fill analysis was used in cases where significant publication bias was detectable. Moreover, we performed subgroup analyses stratified by the type of medication (ACEIs, ARBs, and BBs) and the combination therapy.

## 3. Results and Discussion

### 3.1. Search Results

We identified 2.293 potentially relevant studies from MEDLINE (n = 274), EMBASE (n = 701) and Cochrane Library (n = 1318) search. According to the flow chart (Figure 1) a total of 674 studies were excluded before screening as duplicates records. Of the remaining studies, 1272 were excluded, after screening the title and abstract, as irrelevant to the topic studied in this systematic review. We retrieved the full paper of the remaining 347 studies for detailed evaluation. 339 studies were also excluded due to wrong design and/or wrong population. No trial was identified with additional search strategies. The identification, screening, and inclusion process is illustrated in Figure 1. Overall, eight studies were included in the systematic review and 6 in the meta-analysis conducted. Two studies (Livi et al. [32] and Gulati et al. [33]) were excluded from the meta-analysis. Although those studies were well designed RCT studies, they did not distinguish the results from the HER2 positive patients regarding cardiotoxicity or the changes in LVEF, GLS measurements. Six RCTs were finally included for the analysis with a total of 1056 HER-2 positive breast cancer patients treated with trastuzumab.

The risk of bias of the included studies were assessed using the RoB2 tool [30]. Five out of the eight studies were considered low risk of bias and three of the studies were classified as moderate or high risk of bias concerns. The detailed characteristics of the studies are listed in Table 1 and the detailed risk of bias is shown in Figure 2 and Figure 3. All included studies used solely trastuzumab for the management of HER2-positive breast cancer. We included four RCTs for primary outcomes (Esfandbod et al. [34], Guglin et al. [35], Pituskin et al. [36], Boekhout et al. [37]) and six (Esfandbod et al. [34], Farahani et al. [38], Guglin et al. [35], Sherafati et al. [39], Pituskin et al. [36], Boekhout et al. [37]) were included to assess secondary outcomes.

### 3.2. Study and Patient Characteristics

Of the included papers, five studied the effects of a Beta-Blocker, one the effects of ARB, two the effects of ACEIs and none the effect of a combination therapy. In all studies, patients were not previously treated with ACEIs, ARBs and BBs. Seven studies involved non-metastatic breast cancer and one (Sherafati et al. [39]) administered trastuzumab as adjuvant therapy or as a treatment for metastatic cancer. Seven studies (Boekhout et al. [37], Esfandbod et al. [34], Guglin et al. [35], Farahani et al. [38], Sherafati et al. [39], Pituskin et al. [36], Livi et al. [32]) involved HER-2 positive patients with early stage breast cancer. The identification of trastuzumab-induced cardiotoxicity was based on LVEF, GLS assessment performed by echocardiography (2D or 3D), CMR or MUGA. Seven out of eight studies used echocardiography while the study by Pituskin et al. [35] used only CMR parameters. Guglin et al. [35] and Boekhout et al. [37] used MUGA modality in addition to echocardiography.

Unfortunately, the dosages of the preventive pharmaceutical agents were different for each study. There was a wide range of dosages, between small cohorts, and some of them changed during the study. Therefore, there was high heterogeneity and it was impossible to comparatively evaluated the dosage of each of those agents between studies.

### 3.3. Primary and Secondary Outcomes

The mean follow-up of patients ranged between 3 to 24 months. It is well known that a 3-month cycle of trastuzumab therapy is adequate to induce cardiac dysfunction. Hence, each 3-month cycle maintains the risk for cardiotoxicity. Four studies (Esfandbod et al. [34], Guglin et al. [35], Pituskin et al. [36], Boekhout et al. [37]) involving 828 patients reported data on the primary outcome of cardiotoxicity (Figure 4). The overall effect of medications’ administration was not significant in cardiotoxicity prevention compared to controls (OR = 0.92, 95% CI: 0.54 to 1.56, *p* = 0.75). Treatment with either BBs or ACEIs/ARBs did not reduce the likelihood of cardiotoxicity compared to placebo (OR = 0.92, 95% CI: 0.22 to 3.85, *p* = 0.91 & OR = 0.89, 95% CI: 0.44 to 1.80, *p* = 0.75).

Only two studies used the discontinuation of trastuzumab treatment as secondary outcome (Pituskin et al. [36], Guglin et al. [35]). The use of beta-blockers seemed to lower the likelihood of interruption of trastuzumab therapy than ACEIs. Unfortunately, the lack of adequate number of studies prevented us from conducting analysis and getting a firm conclusion about the efficacy of cardio-protective therapy on the likelihood of trastuzumab interruption.

Six studies including 773 patients assessed the cardioprotective effects of medication based on LVEF data (Figure 5). The increase of LVEF from baseline was significantly higher in patients treated with ACE/ARB inhibitors (MD 2.17%, 95% CI 1.16% to 3.18%, *p* < 0.0001) and to a lesser extent after BBs administration (MD 1.50%, 95% CI −0.24 to 3.25%, *p* = 0.09). There was a high level of heterogeneity overall (I^2^ = 76%, *p* = 0.0001), which was mainly derived by the heterogeneity in the beta-blocker studies (I^2^ = 83%, *p* = 0.0001). At the end of follow-up patients treated with ACEI/ARBs or BBs had slightly higher LVEF than their control counterparts (1.71%, 95% CI 0.50 to 2.92%, *p* = 0.006).

The overall quality assessment of the studies was done applying the GRADE score. All the studies characterized as Moderate or Low grade risk of bias. We didn’t observe any study with Very Low risk. Figure 2 is a summary of the risk of bias.

We identified three studies using serum biomarkers of troponin-T and brain natriuretic peptide (pro-BNP) levels for cardiotoxicity assessment. Two of them (PRADA study [40] and Boekhout et al. [37]) failed to show any correlation of biomarkers changes with cardiotoxicity. Another study (Guglin et al. [35]) simply reported cardiac biomarkers without presenting the results. Obviously, no statistical analysis concerning those biomarkers could be conducted.

The present systematic review and meta-analysis included eight RCTs evaluating the cardioprotective effects of beta-blockers, ACEIs or ARBs in patients with breast cancer receiving the potentially cardiotoxic chemotherapy with trastuzumab. Regarding cardiotoxicity as a dichotomous variable, our meta-analysis failed to demonstrate that pro-active treatment with the aforementioned medications could significantly reduce the risk of trastuzumab-related cardiotoxicity in cancer patients. Based on the left ventricular ejection fraction measurements, the administration of ACEIs/ARBs during trastuzumab chemotherapy period prevented its reduction. Weak evidence supported the association of both ACEIs/ARBs and BBs with reduced trastuzumab treatment interruptions.

A number of previous studies have evaluated the prevention of chemotherapy-induced cardiotoxicity in patients with breast cancer. Those cohorts had predominantly received anthracyclines, but some subgroups received either combined or isolated therapy with trastuzumab. Regarding the increasing application of trastuzumab therapy in clinical practice, to our knowledge this is the first systematic review focusing exclusively on the prevention of trastuzumab-induced cardiotoxicity avoiding the mixed effects with other chemotherapeutic agents. Compared to previous meta-analyses, we included RCTs of patients with breast cancer receiving trastuzumab monotherapy and in parallel with cardio-protective medications. For this reason, we excluded the well-designed clinical trial of Livi et al. [32], since it unfortunately did not stratify analysis for the few HER2 positive patients in the SAFE study. From the methodological point of view, we evaluated cardiotoxicity using either a dichotomous variable or the LVEF change as a continuous variable. Another strength of the current meta-analysis was the pooled analysis of LVEF measurements to reduce the impact of heterogeneity between studies.

The present meta-analysis aimed to address the question of whether there is a clinical benefit from the early administration of cardioprotective drugs in parallel to trastuzumab therapy in patients without any previous cardiac dysfunction. The answer is of great importance regarding the high proportion of patients treated with a combination of anthracycline plus trastuzumab who develop either cardiac dysfunction (27%) or symptomatic HF (16%) [41]. The addition of trastuzumab in chemotherapy regimens has significantly increased the survival benefits along with considerable high cardiotoxic potentiality. Those results require validation in other cohorts in order to accurately calculate the cardiac dysfunction risk of trastuzumab in breast cancer patients. In our meta-analysis prophylaxis with ACEIs/ARBs and/or beta-blockers did not yielded into a reduction of cardiotoxicity. While the rate of cardiac events was nominally better in the prophylaxis arm in the overall analysis (OR = 0.92, 95% CI: 0.54 to 1.56, *p* = 0.75), it failed to reach statistical significance. The hypothesis that prophylaxis with ARBs/ACEIs and/or BBs can reduce late cardiac events remains and should be the subject of adequately powered RCTs. Moreover, this meta-analysis failed to distinguish whether any of the aforementioned pharmaceutical agent is more effective in preventing cardiotoxicity from trastuzumab.

Another important finding of this meta-analysis of six RCTs was that the prophylactic treatment with ARBs/ACEIs was associated with a slight but significant maintenance of LVEF compared to placebo group. From the clinical perspective, the observed amount of LVEF increase after concomitant treatment with trastuzumab and ACEIs/ARBs seemed very modest and within the normal variability of LVEF repeated measurements (MD = 2.17, 95% CI 1.16% to 3.18%, *p* < 0.0001), slightly higher LVEF by 1.71%. Hence, it is not clear from the current evidence whether the prophylactic use of AECIs/ARBs could prevent trastuzumab-induced cardiotoxicity by suppressing the decrease of the echocardiographic-based LV function. Presumably, more studies will verify the observed maintenance of LVEF as an index of cardiotoxicity prevention.

In addition to this, two RCTs documented a 50% reduction of the likelihood of trastuzumab interruption when ARBs/ACEIs and/or BBs were concomitantly administered. Trastuzumab interruption comprises the other side of the same coin, which is cardiotoxicity. Thereby, more RCTs are needed before this index can be incorporated in clinical practice. In this case, we did not detect any superiority of ARBs/ACEIs over BBs on the final result. Factorial design analysis was not included in the analysis. This suggests that ACEI/ARBs and BBs may be prescribed concomitantly with trastuzumab therapy, their optimal dose remains unclear. It was impossible to get firm conclusion about the target dose of those agents as prophylactic therapy. Our meta-analysis did not stratify patients’ risk for cardiotoxicity development. So we are unable to test the current recommendations of pre-treatment of high-risk patients according to 2022 ESC guidelines [15] and the American Society of Clinical Oncology guidelines [42].

Biomarkers of cardiac damage, such as troponin T (TnT) and the amino-terminal fragment of brain natriuretic peptide (NT-proBNP), may be useful as early predictors of cardiac dysfunction. The evaluation of these cardiac biomarkers is theoretically an important component for monitoring of patients undergoing trastuzumab chemotherapy, since they provide evidence of even subtle cardiac dysfunction, before it becomes obvious in echocardiography. Thereby, several prospective studies [43,44,45,46,47] have proposed those cardiac biomarkers, as surrogate indices of cardiotoxicity with controversial results. In our systematic review, we attempted to evaluate the impact of ACEIs/ARBs and or BBs on the aforementioned biomarkers. Only two studies were available for analysis

Overall, the primary prevention of cardiotoxicity due to chemotherapy regimens is still at an early stage and requires more and larger clinical trials. Preclinical studies utilizing valid animal models have elucidated novel pathophysiologic mechanisms and unveiled promising results of pharmaceutical interventions with substantial clinical implications. Khan et al. [48] emphasized the crucial role of antioxidant enzymes in preventing cardiotoxicity induced by trastuzumab. They observed significant heart tissue alterations including cellular disruptions and inflammation in the of trastuzumab-treated rats, along with elevated levels of pro-inflammatory cytokines and cardiac marker enzymes. The study also highlighted the role of oxidative stress and the decrease in antioxidant enzymes in trastuzumab-induced cardiotoxicity. Therefore, medications with established cardioprotective actions are necessary against cardiotoxicity. Prevention from cardiotoxicity has been better studied in patients receiving anthracyclines, but their results may not be applicable to patients receiving trastuzumab, because derived from small studies with different pathophysiological mechanisms. Effective preventive medications presumably require better patients’ selection depending on patients’ co-morbidities. For instance, ACEIs/ARBs may be more effective in hypertensive patients, while BBs in patients with previous myocardial infarction. Perhaps the application of these criteria will reduce the number of patients who need to be treated as a preventative approach. Further adequately powered prospective research on the role of neurohormonal system blockade drugs, like ARNIs, or SGLT2s may be proved more effective as preventive measures in patients at risk of cardiotoxicity from trastuzumab.

This systematic review has several limitations. First of all, the heterogeneity of the studies using different definitions of cardiotoxicity, of outcomes or incomplete data, previous and variable exposure to anthracyclines. Although no patients had cardiac dysfunction at baseline, the frequency of prior exposure to anthracyclines varied considerably from 23% (Pituskin et al. [36]) to 100% (Boekhout et al. [37]), questioning whether the impact of trastuzumab therapy on cardiac function is additive or can stand alone. For the assessment of cardiac dysfunction, we used the current criteria proposed by scientific societies and we omitted studies reporting arbitrarily LVEF changes. Unfortunately, the authors of 2 selected RCTs did not reply to our request for sending data and we unfortunately excluded those studies from our meta-analysis. Finally, the assessment of heterogeneity is less reliable in a meta-analysis with fewer than 10 studies. The asymmetric funnel plot (Figure 6) indicates publication bias regarding the number of patients who experienced cardiotoxicity between the control and intervention group and regarding the changes in LVEF between the control and intervention group.

## 4. Conclusions

In this meta-analysis of six randomized trials, prophylactic use of ARBs/ACEIs or BBs did not reduce the risk of trastuzumab-related cardiotoxicity in HER-2 positive breast cancer patients. Less robust evidence demonstrated their beneficial effect in maintaining, LVEF and from systematic review we noticed less trastuzumab interruptions. Future clinical trials aiming at the primary prevention of cardiotoxicity through established and novel heart failure therapies are needed to evaluate this approach in cardio-oncology. The biomarkers assay and novel imaging modalities may assist an early recognition of cardiotoxicity onset and thereby may help a tailored therapy.

## Figures and Tables

**Figure 1 pharmaceuticals-16-00983-f001:**
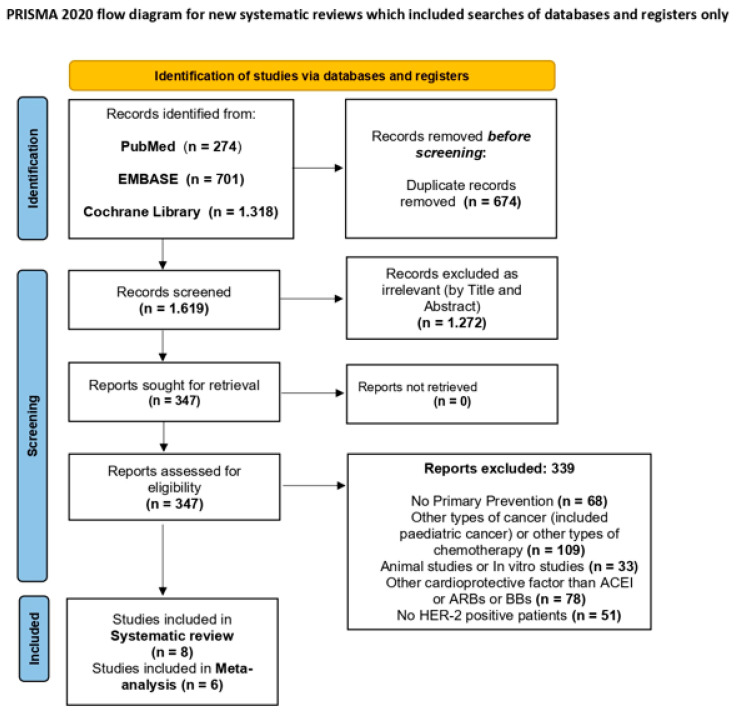
Flow chart for study retrieval and selection. PRISMA 2020. [28] http://www.prisma-statement.org/, accessed on 30 April 2023.

**Figure 2 pharmaceuticals-16-00983-f002:**
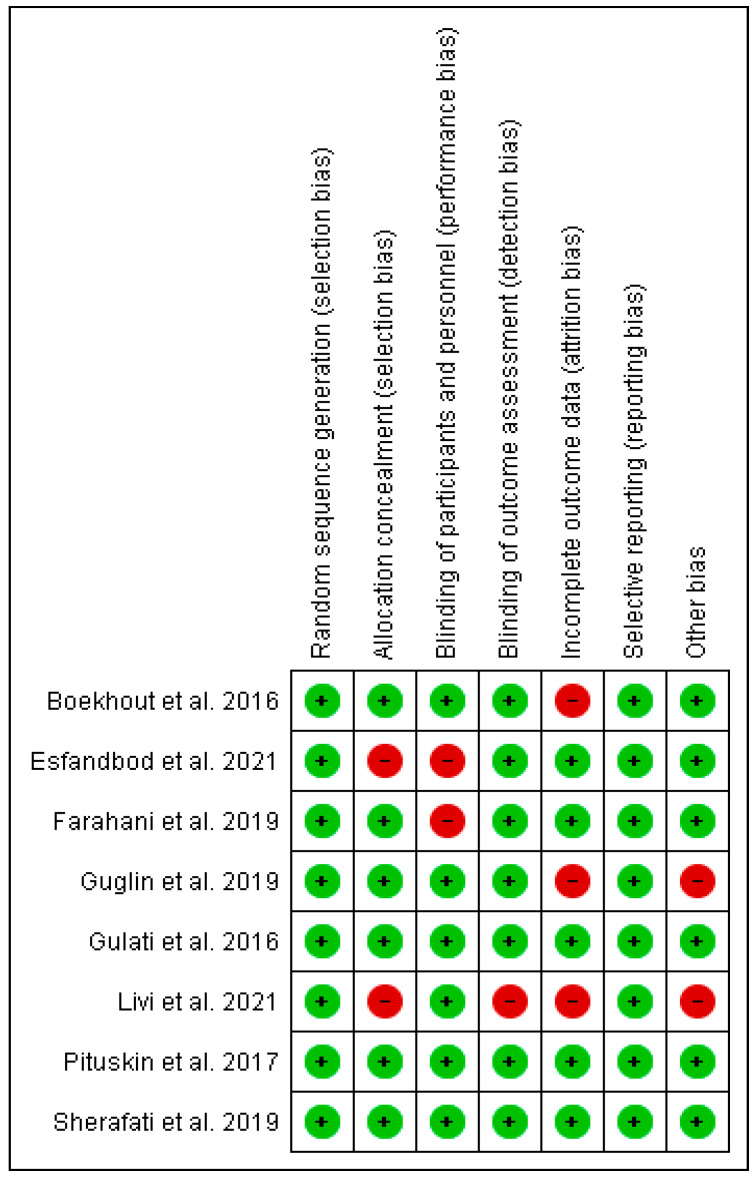
Summary of Risk of Bias. [32,33,34,35,36,37,38,39].

**Figure 3 pharmaceuticals-16-00983-f003:**
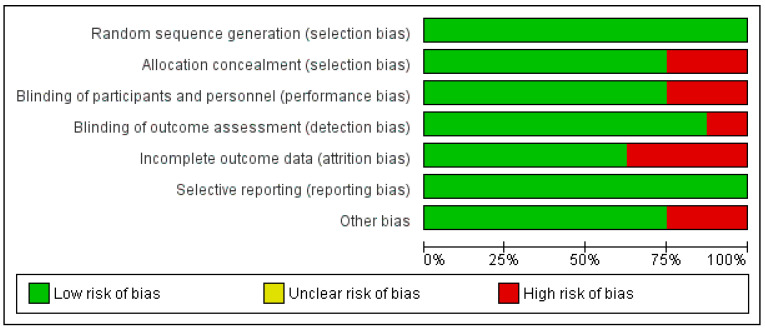
Review author’s judgements about each risk of bias item presented as percent-ages across all included studies.

**Figure 4 pharmaceuticals-16-00983-f004:**
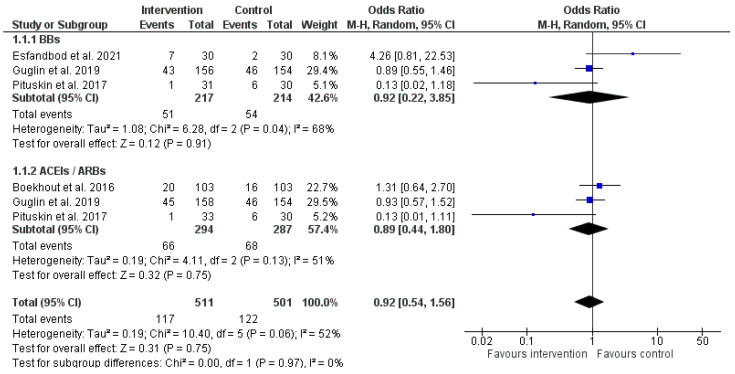
Forest plot illustrating the pooled analysis of the number of patients who led to cardiotoxicity, stratified by drug class. BBs: Beta-Blockers [34,35,36]; ACEIs/ARBs: angiotensin-converting enzyme inhibitors/angiotensin receptor blocker [35,36,37].

**Figure 5 pharmaceuticals-16-00983-f005:**
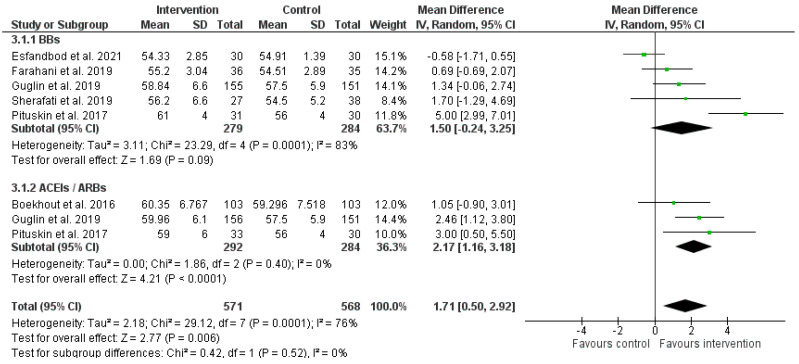
Forest plot illustrating pooled comparison of follow-up LVEF in intervention and control group, stratified by drug class. BBs: Beta-Blockers [34,35,36,38,39]; ACEIs/ARBs: angiotensin-converting enzyme inhibitors/angiotensin receptor blocker [35,36,37].

**Figure 6 pharmaceuticals-16-00983-f006:**
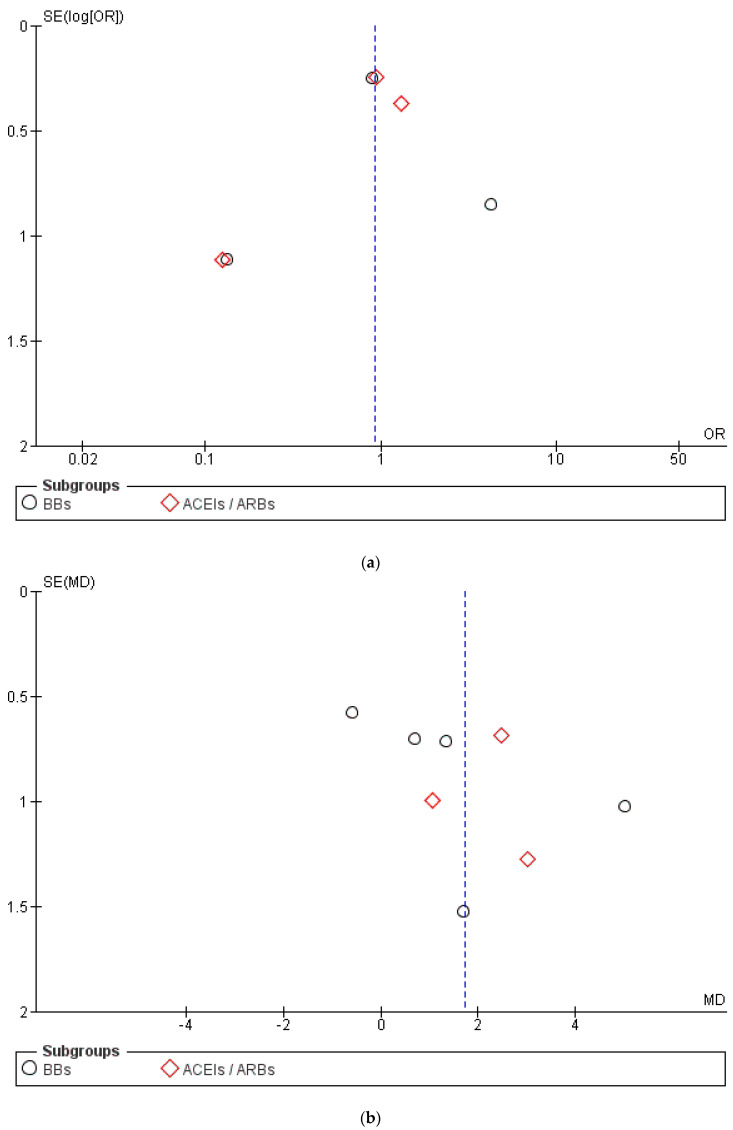
Funnel plot to visualize the publication and other bias: (**a**) The asymmetric funnel plot indicates publication bias regarding the number of patients who experienced cardiotoxicity between the control and intervention group; (**b**) the asymmetric funnel plot indicates publication bias regarding the changes in LVEF between the control and intervention group.

**Table 1 pharmaceuticals-16-00983-t001:** Characteristics and main findings of included studies.

Study ID	Type of Study	Sample Size	Age, Years	Hormonal Receptors Positive	Trastuzumab without Previous Use of Anthracyclines	Adjuvant Trastuzumab	Cardio Protective Agent	Dose of Medication	Type of Baseline Measurements	Duration of Follow-Up	Serum Biomarkers
Livi et al., 2021 [32]	Double-blind, placebo-controlled RCT	174	Median age 48 years	132	_	64	Bisoprolol, Ramipril, or both	5 mg Bisoprolol and/or 5 mg of Ramipril	Standard and 3D echocardiography	24 months	No
Gulati et al., 2016 [33]	Double-blind, placebo-controlled RCT	120	Mean age 50.7 years	_	_	28	Candesartan cilexetil and Metoprolol succinate	Starting dose for candesartan cilexetil was 8 mg and for metoprolol succinate 50 mg, target dose 32 and 100 mg, respectively	CMR and Echocardiography	No follow-up information beyond the adjuvant therapy period	Cardiac troponin I and B-type natriuretic peptide (BNP)
Esfandbod et al., 2021 [34]	Simple RCT	60	Mean age 47 years	60 non-metastatic Her-2 positive patients	_	60	Carvedilol	The dose has been increased in a three-week period to reach 12.5 mg twice a day and continued until the end of therapy.	Echocardiography	12 months	No
Guglin et al., 2019 [35]	Double-blind, placebo-controlled RCT	468	Mean age 51 years	468 HER-2 positive	279	468	Lisinopril, Carvedilol	10 mg once daily	MUGA	12 months	Troponin I and B-type natriuretic peptide (BNP)
Pituskin et al., 2017 [36]	Double-blind, placebo-controlled RCT	94	Mean age 51.3 years	94 HER-2 positive	_	94	Perindopril, Bisoprolol	Daily target doses of Perindopril 8 mg, Bisoprolol 10 mg after was initiated with Perindopril 2 mg daily and Bisoprolol 2.5 mg daily.	CMR	24 months	No
Boekhout et al., 2016 [37]	Double-blind, placebo-controlled RCT	206	Mean age 49.5 years	206 HER-2 positive	_	206	Candesartan	32 mg daily	Echocardiography or MUGA	The median follow-up was 21 months	NT-proBNP and hs-TnT
Farahani et al., 2019 [38]	Open-label RCT	71	Mean age 57 years	71 HER-2 positive	_	71	Carvedilol	6.25 mg twice a day, and 6.25 mg was added to each serving every week to the maximum tolerated dose (12.5 ± 3.125 mg twice a day)	2DSTE	3 months	No
Sherafati et al., 2019 [39]	Double-blind, usual care-controlled RCT	65	Mean age 46.5 years	65 HER-2 positive	_	65	Carvedilol	6.25 mg twice daily	Echocardiography	3 months	No
8		1258			279	1056					

Abbreviations: 2DSTE, 2D speckle-tracking echocardiography; MUGA, Echocardiography and multi-gated acquisition; CMR, cardiac magnetic resonance.

## Data Availability

Data sharing not applicable.

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
