# Peer review of "Cardioprotective Agents for the Primary Prevention of Trastuzumab-Associated Cardiotoxicity: A Systematic Review and Meta-Analysis"

_pharmaceuticals, 2023, doi:10.3390/ph16070983_

Round 1

Reviewer 1 Report

Comments and Suggestion

Generally, the topic is interesting and fits into the scope of the journal. However, some parts of the manuscript should be improved by following journal authors guideline.  For example, Introduction part should little more elaborated and objective should be included in introduction part only.  The Results section should contain the Figure 6 inside the text. Also add the table containing previous finding based on comparative evaluation of ACEIs, 68 ARBs and BBs in cardiotoxicity prevention. Write the  abbreviation of short form atleast one time. In discussion authors should include latest updates in transzumab mediated cardiotoxicity and its prevention by citing khan et al 2023 (Trastuzumab-Mediated Cardiotoxicity and Its Preventive Intervention by Zingerone through Antioxidant and Inflammatory Pathway in Rats). Authors contribution: add the remaining authors like DF and AC

References: Correct the references according to journal guide line. 

Author Response

1st Reviewer

Generally, the topic is interesting and fits into the scope of the journal. However, some parts of the manuscript should be improved by following journal authors guideline.  For example, Introduction part should little more elaborated and objective should be included in introduction part only.  The Results section should contain the Figure 6 inside the text. Also add the table containing previous finding based on comparative evaluation of ACEIs,  ARBs and BBs in cardiotoxicity prevention. Write the  abbreviation of short form at least one time.

We would like to thank reviewer for the suggestion to elaborate the introduction by providing more data about the hypothesis of preventive effects of ACEIs/ARBs, BBs. We have also moved the objective of the study in the introduction part only.

Following reviewer’s suggestion, we have moved Figure 6 inside the text in the Results section.

We have also mentioned all abbreviations at first sight.

In discussion authors should include latest updates in trastuzumab mediated cardiotoxicity and its prevention by citing khan et al 2023 (Trastuzumab-Mediated Cardiotoxicity and Its Preventive Intervention by Zingerone through Antioxidant and Inflammatory Pathway in Rats). Authors contribution: add the remaining authors like DF and AC

Thanks’ to reviewer’s suggestion we have updated the discussion by adding the proposed reference. The contribution of all authors is mentioned at the end of the manuscript.

Correct the references according to journal guide line

All references are corrected.

Reviewer 2 Report

Manuscript Number: pharmaceuticals-2410328

The manuscript entitled “Cardioprotective Agents for the Primary Prevention of Trastuzumab-Associated Cardiotoxicity: A Systematic Review and Meta-Analysis” is well narrated. The data are well presented. This study report is very interesting and suitable for this journal.

Comments

1.    Introduction part you can start with cardiotoxicity or drug-induced cardiotoxicity?

2.    All the patients are in the same age group?

3.    The lane 53, “From the methodological point of view, all …, please simplify the sentence.

4.    In lane 56, “Perhaps a possible rea- 56

 son for the limited number of publications…. You can modify the sentence or delete it.

5.    For analysis how you select the data and any specific tool for analyzing the data?

6.    In lane 141, please write the full form for LVEF or GLS

7.    What do you mean by “339 studies were also excluded due to wrong design 168 and/or wrong population.?

8.    You mentioned that “Unfortunately, the dosages of the preventive pharmaceutical agents were different for each study’’. In this case, do you consider any dose range selection for meta-analysis?

9.    In Figure 1, “Records removed before screening”, Duplicate records were removed (n = 674). What are the criteria for considering a duplicate record? Like the same dosage or duration?

10. In lane 213 The mean follow-up of patients ranged between 3 to 24 months. This is a large range of patients, do you have any subcategorization to find out 828 patients reported data on the primary outcome of cardiotoxicity?

11. Cardiotoxicity-related serum biomarkers more prominently used also you can illustrate in a pic/graph representation?

12. In Table 1 , make everything in a uniform manner and aligned properly without breaking the words.

13. In lane345, Effective preventive medications presumably require better patient selection depending on patients’ co-morbidities”. What do you mean here?

14. You can add correlation analysis with different data or biomarkers?

15. Discussion and conclusions were well written.

Author Response

2nd Reviewer

Comments

  1. Introduction part you can start with cardiotoxicity or drug-induced cardiotoxicity?

We really appreciate reviewer’s suggestion. This is a systematic review / meta-analysis and we preferred to follow PICO framework. For this reason, we firstly describe the under-investigation population, then the intervention and then the drug-induced cardiotoxicity. We believe it is essential to follow PICO principles throughout the manuscript.

  1. All the patients are in the same age group?

3.1, 3rd paragraph: This is a useful comment. We have now clarified that all patients were in the same age group between 30-70 years old.

  1. The lane 53, “From the methodological point of view, all …, please simplify the sentence.

We have simplified the sentence

  1. In lane 56, “Perhaps a possible reason for the limited number of publications…. You can modify the sentence or delete it.

We appreciate reviewer’s suggestion, We have modified the sentence.

  1. For analysis how you select the data and any specific tool for analyzing the data?

We have used the RevMan version 5.4.1.software for data collection and analysis which is mentioned in statistical analysis section.

  1. In lane 141, please write the full form for LVEF or GLS

We have explained the abbreviations already in lines 37 and 69.

  1. What do you mean by “339 studies were also excluded due to wrong design 168 and/or wrong population.?

Wrong design means they were not randomized trials including observational, study cohorts etc or the patients received at the end mixed chemotherapies.

Wrong population means that the patients did not have exclusively HER2 positive breast cancer, or there were mixed populations where it was not clarified the type of cancer 

  1. You mentioned that “Unfortunately, the dosages of the preventive pharmaceutical agents were different for each study’’. In this case, do you consider any dose range selection for meta-analysis?

Lines 199-201: This is an important comment. There was a wide range of dosages, in small cohorts, and some of them changed during the study. So our meta-analysis was underpowered to perform any related meta-analysis of the dosage.

  1. In Figure 1, “Records removed before screening”, Duplicate records were removed (n = 674). What are the criteria for considering a duplicate record? Like the same dosage or duration?

Duplicate records are considered identical records retrieved from multiple databases or multiple articles published from the same dataset.

  1. In lane 213 The mean follow-up of patients ranged between 3 to 24 months. This is a large range of patients, do you have any subcategorization to find out 828 patients reported data on the primary outcome of cardiotoxicity?

We really thank the reviewer for this excellent comment which points out the heterogeneity between studies. It is well known that a 3-month cycle of trastuzumab therapy is adequate to induce cardiac dysfunction. Hence, each 3-month cycle maintains the risk for cardiotoxicity.

  1. Cardiotoxicity-related serum biomarkers more prominently used also you can illustrate in a pic/graph representation?

Although our initial hypothesis was the significant contribution of biomarkers to cardiotoxicity detection and monitoring, we identified only 3 studies using different biomarkers. That prevented us from drawing firm conclusion about their usage in clinical practice in this certain population treated exclusively with trastuzumab.

  1. In Table 1 , make everything in a uniform manner and aligned properly without breaking the words.

Thanks reviewer for the suggestion.

  1. In lane345, Effective preventive medications presumably require better patient selection depending on patients’ co-morbidities”. What do you mean here?

We have added a sentence explaining that statement.

  1. You can add correlation analysis with different data or biomarkers?

We really appreciate reviewer’s suggestion. This is a systematic review and meta-analysis and we followed the principles of this analysis, where correlation analysis will not add more information.

  1. Discussion and conclusions were well written.

Reviewer 3 Report

The first sentence of your abstract does not jive with what you have found. Consider changing it to “The perception of …” or “considerations of toxicity….”

In your introduction you note that “The incidence of trastuzumab-induced cardiotoxicity may range between 5 and 11% of treated patients, but there are no available indices which could reliably predict it.” You might consider noting that toxicity has been reported to range…. These numbers, although they are in the literature, may (or may not) be way higher than what really exists, especially when trastuzumab is given without other agents that affect toxicity.

In your discussion you mention the unfortunate numbers in the Slamon paper (your reference 40), that have never been duplicated, and that even at the time of publication were considered by some to be outliers.

Author Response

3rd Reviewer

The first sentence of your abstract does not jive with what you have found. Consider changing it to “The perception of …” or “considerations of toxicity….”

Following reviewer’s suggestion we have rephrased the first sentence in the abstract

In your introduction you note that “The incidence of trastuzumab-induced cardiotoxicity may range between 5 and 11% of treated patients, but there are no available indices which could reliably predict it.” You might consider noting that toxicity has been reported to range…. These numbers, although they are in the literature, may (or may not) be way higher than what really exists, especially when trastuzumab is given without other agents that affect toxicity.

We fully agree may differ in clinical practice depending on the cohorts co-morbidities or the way of monitoring. The related comment has been added

In your discussion you mention the unfortunate numbers in the Slamon paper (your reference 40), that have never been duplicated, and that even at the time of publication were considered by some to be outliers.

Thank you very much for the comment. It has been noted that the results of this study require further validation.